# Long-Term Effectiveness and Sustainability of Integrating Peer-Assisted Ultrasound Courses into Medical School—A Prospective Study

**Johannes Matthias Weimer** [1,*], **Nina Widmer** [1], **Kai-Uwe Strelow** [1,*], **Paula Hopf** [1], **Holger Buggenhagen** [1], **Klaus Dirks** [2], **Julian Künzel** [3], **Norbert Börner** [4], **Andreas Michael Weimer** [5], **Liv Annebritt Lorenz** [6], **Maximilian Rink** [3], **Henrik Bellhäuser** [7], **Lina Judit Schiestl** [8], **Roman Kloeckner** [9], **Lukas Müller** [10,†] **and Julia Weinmann-Menke** [11,†]

1   Rudolf Frey Learning Clinic, University Medical Centre of the Johannes Gutenberg-University Mainz, 55131 Mainz, Germany; nwidmer@students.uni-mainz.de (N.W.); paulhopf@uni-mainz.de (P.H.); buggenha@uni-mainz.de (H.B.)
2   Department of General Internal Medicine and Geriatrics, Rems-Murr-Klinikum, 71364 Winnenden, Germany; klaus.dirks@rems-murr-kliniken.de
3   Department of Otorhinolaryngology, Head and Neck Surgery, University Hospital Regensburg, 95053 Regensburg, Germany; julian.kuenzel@klinik.uni-regensburg.de (J.K.); maximilian.rink@klinik.uni-regensburg.de (M.R.)
4   Gastroenterological Medical Group Offices at the MED Specialist Centre Mainz, 55131 Mainz, Germany; noboerner@gmail.com
5   Centre of Orthopaedics, Trauma Surgery and Spinal Cord Injury, Heidelberg University Hospital Heidelberg, 69118 Heidelberg, Germany
6   Department of Radiation Oncology and Radiotherapy, University Medical Centre of the Johannes Gutenberg-University Mainz, 55131 Mainz, Germany; liv-annebritt.lorenz@unimedizin-mainz.de
7   Institute of Psychology, Johannes Gutenberg University of Mainz, 55112 Mainz, Germany; bellhaeuser@uni-mainz.de
8   Department of Gynaecology and Obstetrics, University Medical Centre of the Johannes-Gutenberg University Mainz, 55131 Mainz, Germany; lina.schiestl@unimedizin-mainz.de
9   Institute of Interventional Radiology, University Hospital Schleswig-Holstein—Campus Lübeck, 23583 Lübeck, Germany; roman.kloeckner@uksh.de
10   Department of Diagnostic and Interventional Radiology, University Medical Centre of the Johannes Gutenberg-University Mainz, 55131 Mainz, Germany; lukas.mueller@unimedizin-mainz.de
11   I. Department of Medicine, University Medical Centre of the Johannes Gutenberg University-Mainz, 55131 Mainz, Germany; julia.weinmann-menke@unimedizin-mainz.de
*   Correspondence: weimer@uni-mainz.de (J.M.W.); kai-uwe.strelow@uni-mainz.de (K.-U.S.)
†   These authors contributed equally to this work.

**Abstract:** Introduction: Ultrasound diagnostics is an important examination method in everyday clinical practice, but student education is often inadequate for acquiring sufficient basic skills. Individual universities have therefore started integrating (extra)curricular training concepts into medical education. This study aimed to evaluate sustainable skills development through participation in peer-assisted ultrasound courses. Methods: From 2017, students in the clinical part of medical school could opt for extracurricular peer-assisted ultrasound courses. Depending on the format (10-week course/2-day compact course) these comprised 20 teaching units focusing on abdominal and emergency ultrasonography. Students attending compulsory workshops at the start of their practical year were enrolled in this study, allowing for a comparison between the study group (attended ultrasound course) and the control group (did not attend ultrasound course). Competency from two out of four practical exams (subjects: "aorta", "gallbladder", "kidney" and "lung") was measured, and a theory test on the same subject areas ("pathology recognition") was administered. Additional questions concerned biographical data, subjective competency assessment (7-point Likert scale), and "attitude to ultrasound training in the curriculum". Results: Analysis included 302 participants in total. Ultrasound courses had been attended on average 2.5 years earlier (10-week course) and 12 months earlier (2-day compact course), respectively. The study group ($n = 141$) achieved significantly better results than the control group ($n = 161$) in the long-term follow-up. This applies both to

practical exams ($p < 0.01$) and theory tests ($p < 0.01$). After course attendance, participants reported a significantly higher subjective assessment of theoretical ($p < 0.01$) and practical ($p < 0.01$) ultrasound skills. Conclusions: Peer-assisted ultrasound courses can sustainably increase both theoretical and practical competency of medical students. This highlights the potential and need for standardised implementation of ultrasound courses in the medical education curriculum.

**Keywords:** ultrasound diagnostics; sonography; medical education; ultrasound curriculum; peer-assisted learning; course models; undergraduate training; curriculum development

## 1. Introduction

### 1.1. Background

Imaging modalities such as computed tomography (CT), magnetic resonance imaging (MRI), X-ray, and ultrasound are important diagnostic tools in everyday clinical practice and can be used in many ways for differential diagnosis. Of these imaging modalities, ultrasound has become increasingly important due to its lack of radiation exposure and its ability to be used anywhere, and is now firmly established in a number of disciplines [1]. This is why profound and early education in this examination method is so important. Imparting clinical practical and theoretical skills is a key aspect of medical education in preparing students for their eventual work as clinicians. In that respect, ongoing modifications and refinements are taking place as part of curricular development in order to achieve a sustainable skills build-up [2]. Some preclinical and clinical training in medical schools already incorporates ultrasound-specific education in the form of (peer-assisted) elective and compulsory courses [3–10]. Furthermore, existing recommendations by international professional associations foster implementation with regard to teaching staff, teaching methods, teaching material, interactivity, motivation, and resource management [11–14]. Recently published training approaches employ teaching formats that last several weeks or throughout a semester, but also compact courses, sometimes across overlapping semesters [10,15–25]. Measurement of competency in these formats usually involves theory tests, practical exams, and evaluation forms [26]. Not only the short-term gain in competency, but also long-term learning success, is an important aspect, and the subject of current research [27–33].

### 1.2. Research Problem and Question

Since the summer semester of 2017, peer-assisted ultrasound-specific training formats in the form of an elective course have been integrated into the clinical part of medical studies at the University Medical Centre Mainz. These comprised at least 20 teaching units depending on the format (10-week course or 2-day compact course) [34]. Abdominal and emergency ultrasound are the chosen subjects of the learning content, based on the learning objective catalogue of the German Society of Ultrasound in Medicine (DEGUM) [35]. These two sub-areas were chosen because they are among the most common applications of clinical sonography and are performed by many disciplines. The extent to which sustainable, long-term gain in competency in ultrasound diagnostics is achievable through additional ultrasound training formats has so far remained unanswered from the perspective of educational research. Therefore, the following study aims to investigate whether the integration of ultrasound-specific training programmes in the (early) clinical part of medical studies leads to sustainable development of competency among participants. In particular, we aim to evaluate and measure subjectively and objectively the gain of practical as well as theoretical skills. Auxiliary questions relate to students' general attitude to, and acceptance of, restructuring the training curricula.

## 2. Materials and Methods

### 2.1. Study Design, (Recruitment of) Participants and Study Procedure

This single-centre prospective observation study (see Figure 1) took place at the Rudolf Frey Learning Clinic at Mainz University Medical Centre in cooperation with the Department of Diagnostic and Interventional Radiology and the I. Department of Medicine. Its implementation complied with the Strengthening the Reporting of Observational Studies in Epidemiology STROBE criteria [36]. Inclusion criteria were defined as completion of the written tests and participation in the practical examination, and the agreement to participate in this study. This study included medical students from the winter semester 2019/2020 and the summer semester 2021 at the start of their practical year as part of centrally organised compulsory workshops for practical year students in the federal state of Rhineland-Palatinate. Generally speaking, the clinical, curricular education of all the participants up to the relevant data collection period can be deemed equivalent as there were no fundamental changes to the running and content of the medical degree programme at Mainz University Medical Centre. The extracurricular education can vary depending on the additional courses attended. As part of the workshop the participants completed an evaluation, a theory test, and two practical ultrasound exams [26]. The data collection method was equivalent within the workshops.

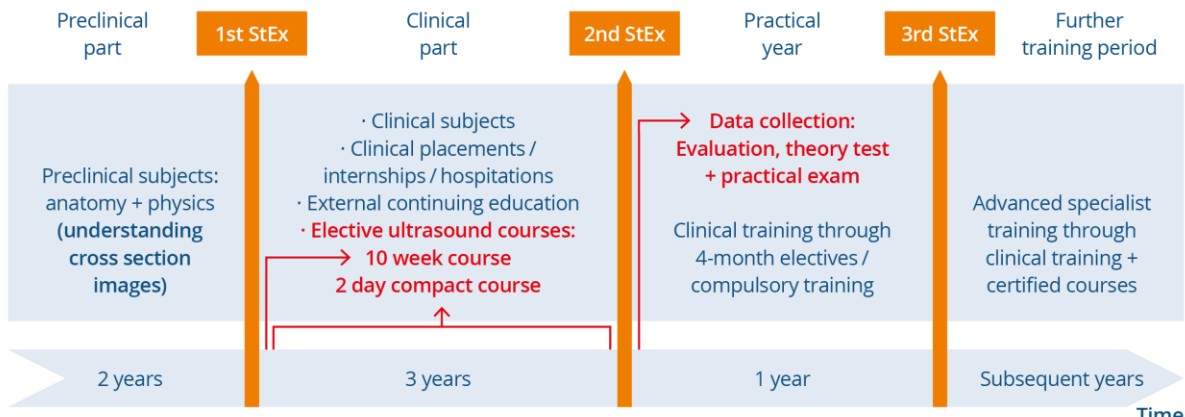

**Figure 1.** This illustration shows the current time plan of medical studies at the Johannes Gutenberg University Mainz, which consists of the pre-clinical study period, the clinical study period, and the practical year. Possible points of contact with ultrasound and data collection time in this study are listed. Participants could only attend the 10-week course in the first clinical semester. They could attend the 2-day compact course throughout their clinical training phase. Data collection took place after the 2nd State Exam/at the start of the practical year. The figure also shows possible further training opportunities after completing studies/beginning specialist training.

We then formed two groups from the total group of all participants: the study group (*n* = 161) who attended at least one extracurricular ultrasound course during their clinical training, and the control group (*n* = 141) who attended no ultrasound courses during their studies. There were three subgroups defined within the study group ("10-week course" (*n* = 90), "2-day compact course" (*n* = 39), "both courses" (*n* = 12)). Supplement Figure S1 presents the process of enrolment, allocation, and analysis in the form of a flow chart diagram (in accordance with CONSORT).

### 2.2. Measuring Instruments

#### 2.2.1. Evaluation

By using specific items, the evaluation form asked about various areas. These included "personal data", "education/study/profession", "attendance at ultrasound courses", "previous ultrasound experience", "subjective assessment of competency", and "satisfaction,

expectations, and need". We used dichotomous questions ("yes/no"), single and multiple-choice answers, free text answers, and seven-point Likert scales.

2.2.2. Written Test

The theory test (see Supplementary Figure S2) comprised four ultrasound pathology pictures for participants to label (aortic aneurysm, cholecystitis, renal congestion, pleural effusion). Participants had eight minutes to answer the questions (max. 4 assessment points (AP)).

2.2.3. Practical Examination

We modified the test sheets of Hofer, et al. [37] on the subjects "aorta", "gallbladder", and "kidney" and used them for the practical ultrasound exam, and we created another test on the subject "pleura" (max. number of points per exam 50). The areas assessed in the examination were "opening the dialogue" (4 AP), "handling the transducer" (8 AP), "patient guidance" (6 AP), "examination procedure" (13 AP), "overall performance" (8 AP), "communication" (4 AP), and "theory questions" (7 AP) (see Supplementary 3). Our definition of the skills expected in the practical exams under "opening the dialogue", "patient guidance", and "communication" was derived from Jünger, et al. [38].

We randomly allocated the participants to two of the four exams. Participants had 8 min to perform each exam. Before entering the examination room, they received a case vignette with an introduction to the scenario as well as instructions on how to carry out the exam (4 min). In the role of the clinician they then had to perform an ultrasound examination on an actor patient based on the clinical picture being questioned in the case vignette. The actor patients (volunteer students) had a similar body mass index and were asked to come to the examination fasting. Previously trained peer tutors supervised the examination and the assessment. The training consisted of a multi-stage process that included participation in a DEGUM-certified ultrasound course, internal technical and pedagogical training (30 h), and shadowing in the ultrasound laboratory (with the performance of at least 100 independent examinations). The acquisition of competence through this multi-stage training programme was verified with a practical examination carried out by the physicians conducting the study. The peer tutors had to demonstrate all the practical, theoretical, and didactic skills required for the course in this trial session. Ultrasound machines from GE HealthCare (GE F8; General Electric Company, Boston, MA, USA) and Philips (HD 5; Philips GmbH, Hamburg, Germany), each with high-frequency linear transducers and presets specially configured for the course, were used for both training and examination.

*2.3. Statistical Analysis*

We conducted all the statistical analyses and produced all the graphics using R studio (RStudio Team, RStudio: Integrated Development for R. 2020) (Posit Software, Vienna, Austria) with R 4.0.3 (R Foundation for Statistical Computing, A Language and Environment for Statistical Computing) (Posit Software, Vienna, Austria)Binary and categorical baseline parameters are expressed as absolute numbers and percentages. Continuous data are expressed as median and interquartile range (IQR), or as mean and standard deviation (SD). We tested data distribution using the Shapiro–Wilk test. As the variables were not normally distributed, we compared categorical parameters using Fisher's exact test and continuous parameters using the Mann–Whitney test. Furthermore, we built a multivariate linear regression model to compare the influence of individual factors to the results of the practical exam score. All $p$ values < 0.05 were considered statistically significant.

**3. Results**

A total of 319 students were scanned for participation. $N = 3$ participants had incomplete data sets and $n = 14$ participants declined to participate and were therefore excluded. The statistical analysis included a total of $n = 302$ complete data sets (study group with

*n* = 161 and control group with *n* = 141). If ultrasound courses had been attended (study group), these were an average of 2.5 years ago (10-week course) or 12 months ago (2-day compact course). Table 1 lists the participants' demographic details. In this respect, we found no statistical differences in the groups analysed. The majority stated that they had come into contact with ultrasound during their clinical placements ("Famulaturen") (a total of 263 participants or >85.00% per group). Most of the students had independently examined fewer than 51 patients.

**Table 1.** Characteristics of the participants (*n* = 302).

| Variables | Control Group (*n* = 161) | Study Group (*n* = 141) | *p* Value |
|---|---|---|---|
| Attended weekend course, n (%) | 0 (0.0) | 39 (27.6) | |
| Attended weekly course, n (%) | 0 (0.0) | 90 (63.8) | |
| Attended weekend course + weekly course, n (%) | 0 (0.0) | 12 (8.6) | |
| Age, mean (SD) | 28.4 (3.6) | 27.5 (3.1) | 0.05 |
| Gender | n (%) | n (%) | 0.30 |
| female | 83 (51.6) | 82 (58.2) | |
| male | 77 (47.8) | 59 (41.8) | |
| no information | 1 (0.6) | 0 (0.0) | |
| Training prior to study | n (%) | n (%) | 0.64 |
| yes | 88 (54.7) | 81 (57.5) | |
| no | 73 (45.3) | 60 (42.5) | |
| Previous medical experience | n (%) | n (%) | 0.99 |
| yes | 92 (57.1) | 81 (57.5) | |
| no | 69 (42.9) | 60 (42.5) | |
| Contact points with ultrasound in clinical placement | n (%) | n (%) | 0.60 |
| no | 18 (11.2) | 19 (13.5) | |
| yes | 142 (88.2) | 121 (85.8) | |
| no information | 1 (0.6) | 1 (0.7) | |
| Internal medicine (hospital) | 80 (49.7) | 63 (44.7) | |
| GP | 93 (57.8) | 95 (67.4) | |
| Radiology | 5 (3.1) | 10 (7.1) | |
| Gynaecology | 19 (11.8) | 13 (9.2) | |
| Urology | 21 (13.0) | 4 (2.8) | |
| Paediatrics | 0 (0.0) | 0 (0.0) | |
| Anaesthesiology | 0 (0.0) | 0 (0.0) | |
| Surgery | 0 (0.0) | 0 (0.0) | |
| Number of patients independently examined | n (%) | n (%) | 0.66 |
| 0–50 | 132 (82.0) | 114 (80.9) | |
| 50–100 | 10 (6.2) | 12 (8.5) | |
| 100–150 | 1 (0.6) | 2 (1.4) | |
| 150–200 | 0 (0.0) | 0 (0.0) | |
| >200 | 1 (0.6) | 2 (1.4) | |
| No information | 17 (10.6) | 11 (7.8) | |
| Other ultrasound experience | n (%) | n (%) | 0.32 |
| no | 134 (83.2) | 122 (86.5) | |
| yes | 25 (15.5) | 16 (11.4) | |
| no information | 2 (1.2) | 3 (2.1) | |

*3.1. Results of Evaluations*

Table 2 presents the evaluation results of the question regarding participants' "expectations and need" and "subjective assessment of competency". The students, irrespective of their group, were in favour of "integration of ultrasound training into compulsory teaching during their studies" (mean 1.45 [SD 0.98] SP (scale points), respectively, mean 1.41 [SD 0.90] SP; *p* = 0.91) and would like "further development of digital teaching media" (mean 1.88 [SD 1.22] SP, respectively, mean 1.70 [SD 1.03] SP; *p* = 0.16).

**Table 2.** Self-assessment, expectation, and examination results of the participants.

| Variables | Control Group (*n* = 161) | Study Group (*n* = 141) | *p* Value |
|---|---|---|---|
| | Mean (SD) | Mean (SD) | |
| **Expectations and need (1 = fully agree; 7 = do not agree at all)** | | | |
| Diagnostic competency during studies | 1.38 (0.91) | 1.38 (1.04) | 0.59 |
| Diagnostic competency within compulsory teaching | 1.45 (0.98) | 1.41 (0.90) | 0.91 |
| Integration of digital teaching media into ultrasound education | 2.04 (1.35) | 1.81 (1.09) | 0.10 |
| Further development of digital teaching media | 1.88 (1.22) | 1.70 (1.03) | 0.16 |
| **Current subjective assessment of competency (1 = very low, 7 = very high)** | | | |
| Theoretical ultrasound knowledge | 2.88 (1.22) | 3.39 (1.36) | <0.01 |
| Practical ultrasound knowledge | 2.70 (1.27) | 3.36 (1.31) | <0.01 |
| Topographical anatomical knowledge | 3.69 (1.28) | 4.11 (1.35) | 0.01 |
| Spatial perception/orientation in the image | 3.61 (1.36) | 4.16 (1.4) | <0.01 |
| Handling of an ultrasound machine | 3.57 (1.24) | 4.38 (1.39) | <0.01 |
| Optimal adjustment of the image | 2.79 (1.34) | 3.51 (1.36) | <0.01 |
| Retrievable knowledge from that time | | % (SD) | |
|    Two-day compact course (*n* = 39) | 0 | 48.2 (21.5) | |
|    10-week course (*n* =90) | 0 | 42.9 (17.7) | |
|    Both courses (*n*= 12) | 0 | 72.05 (22.6) | |

The study group's assessment of their "current competencies" in all the skill areas addressed was significantly higher than that of the control group (*p* < 0.01). This applies to theoretical skills (mean 2.88 [SD 1.22] SP vs. mean 3.39 [SD 1.36] SP) as well as practical skills (mean 2.70 [SD 1.27] SP vs. MW 3.36 [SD 1.31] SP), as reflected tendentially in the subgroup analysis (see Supplementary Figure S3).

Students who had participated in both course formats assessed their "knowledge at that time", which they were additionally asked about, better than those who had only attended one course format (72.05% SD [22.6] % both course formats vs. 48.2% SD [21.5] % 10-week course vs. 42.9% SD [17.7] % 2-day compact course).

### 3.2. Results of Theory Tests and Practical Examinations

Table 3 and Figure 2 present the results of the theory tests and the practical examinations. The students in the study group achieved significantly higher results (*p* < 0.01) than the students in the control group. This applies both to the theory test (mean 2.8 [SD 1.08] SP vs. mean 2.43 [SD 1.18] SP) and the overall average of the practical exams (mean 33.2 [SD 6.6] SP vs. MW 28.6 [SD7.75] SP).

**Table 3.** Results of written and practical exams.

| | Control Group | Study Group | *p* Value |
|---|---|---|---|
| | Mean (SD) | Mean (SD) | |
| **Overall test and examination results** | | | |
| Pathology diagnoses (theory test) (max. 4 AP) | 2.43 (1.18) | 2.8 (1.08) | <0.01 |
| Practical exam average (max. 50 AP) | 28.6 (7.75) | 33.2 (6.6) | <0.01 |
| **Test results per practical exam** | | | |
| **Exam—lung (n)** | 72 | 54 | |
| Total (max. 50 P) | 30.6 (8.30) | 35.3 (7.33) | <0.01 |
| Opening the dialogue (max. 4 P) | 3.32 (0.87) | 3.20 (0.77) | 0.14 |

**Table 3.** *Cont.*

|  | Control Group | Study Group | *p* Value |
|---|---|---|---|
|  | **Mean (SD)** | **Mean (SD)** |  |
| Handling the transducer (max. 8 P) | 4.72 (1.75) | 5.5 (1.56) | <0.01 |
| Patient guidance (max. 6 P) | 2.83 (2.13) | 4.25 (1.86) | <0.01 |
| Examination procedure (max. 13 P) | 8.26 (3.0) | 9.93 (2.12) | <0.01 |
| Overall performance (max. 8 P) | 4.57 (1.73) | 5.35 (1.62) | 0.01 |
| Communication (max. 4 P) | 2.60 (1.21) | 2.55 (1.00) | 0.77 |
| Theory questions (max. 7 P) | 4.32 (1.10) | 4.55 (1.22) | 0.39 |
| **Exam—gallbladder (n)** | 71 | 50 |  |
| Total (max. 50 P) | 28.5 (7.89) | 32.1 (6.65) | <0.01 |
| Opening the dialogue (max. 4 P) | 3.41 (0.74) | 3.10 (0.73) | <0.01 |
| Handling the transducer (max. 8) | 4.78 (1.74) | 5.17 (1.36) | 0.27 |
| Patient guidance (max. 6 P) | 2.89 (2.19) | 4.05 (1.90) | <0.01 |
| Examination procedure (max. 16 P) | 8.68 (3.04) | 10.40 (2.90) | <0.01 |
| Overall performance (max. 8 P) | 4.02 (1.68) | 4.79 (1.39) | <0.01 |
| Communication (max. 4 P) | 2.41 (0.97) | 2.42 (0.87) | 0.93 |
| Theory questions (max. 4 P) | 2.14 (0.94) | 2.19 (0.72) | 0.55 |
| **Exam—aorta (n)** | 60 | 41 |  |
| Total (max. 50 P) | 25.5 (8.20) | 30.70 (8.33) | <0.01 |
| Opening the dialogue (max. 4 P) | 3.27 (0.82) | 2.96 (0.75) | 0.02 |
| Handling the transducer (max. 8 P) | 5.12 (1.76) | 5.95 (1.33) | 0.02 |
| Patient guidance (max. 6 P) | 2.03 (2.13) | 3.05 (2.13) | 0.02 |
| Examination procedure (max. 13 P) | 6.72 (3.09) | 8.60 (2.65) | <0.01 |
| Overall performance (max. 8 P) | 3.98 (1.52) | 4.98 (1.61) | <0.01 |
| Communication (max. 4 P) | 2.24 (0.98) | 2.40 (0.92) | 0.52 |
| Theory questions (max. 7 P) | 2.15 (1.42) | 2.68 (2.00) | 0.35 |
| **Exam—kidney (n)** | 61 | 61 |  |
| Total (max. 50 P) | 32.0 (7.62) | 33.6 (5.92) | 0.18 |
| Opening the dialogue (max 4 P) | 3.19 (0.84) | 3.14 (0.69) | 0.34 |
| Handling the transducer (max. 8 P) | 5.43 (1.67) | 5.80 (1.42) | 0.24 |
| Patient guidance (max. 6 P) | 3.16 (2.20) | 3.57 (2.02) | 0.30 |
| Examination procedure (max. 14 P) | 9.45 (2.56) | 9.98 (2.46) | 0.30 |
| Overall performance (max. 8 P) | 5.11 (1.62) | 5.32 (1.38) | 0.37 |
| Communication (max. 4 P) | 2.48 (0.92) | 2.54 (0.85) | 0.77 |
| Theory question (max. 6 P) | 3.20 (0.87) | 3.26 (0.87) | 0.79 |

Subgroup analysis of the 10-week course (*n* = 90), 2-day compact course (*n* = 39), and both courses (*n* = 12) also confirms these results. Participants who attended both course formats achieved the highest results in the theory and practical tests (theory mean 3.25 [SD 1.14] SP and practical mean 41.6 [SD 8.46] SP), followed by those in the 10-week course (theory mean 2.83 [SD 1.11] SP and practical mean 32.8 [SD 6.02] SP). The results of the participants in the 2-day compact course were slightly lower (theory mean 2.59 [SD 0.97] SP and practical mean 31.7 [SD 5.45] SP).

Separate analysis of the four practical exams and their subitems (Figure 2 and Table 3) further reveals that the study group achieved significantly higher results than the students in the control group in "opening the dialogue" of practical exam 3 and in "patient guidance", "examination procedure", and "overall performance" of practical exams 1 to 3. We found no significant differences in all the exams with regard to "communication during the examination" and "theory questions". In exam four (subject kidney) participants in both groups achieved results without a significant difference in all the subitems.

Multivariate linear regression analysis showed significant effects of "ultrasound course attendance" ($\beta$ = 3.40; $p < 0.001$) and the "number of independently performed ultrasound examinations of more than 50" ($\beta$ = 3.20; $p = 0.04$) on the results of the practical exam score. The items "ultrasound seen/performed during clinical placement" ($\beta$ = 2.72; $p = 0.11$), "previous medical experience" ($\beta$ = 1.28; $p = 0.13$), and "other ultrasound experience" ($\beta$ = 1.33; $p = 0.33$) had no significant influence.

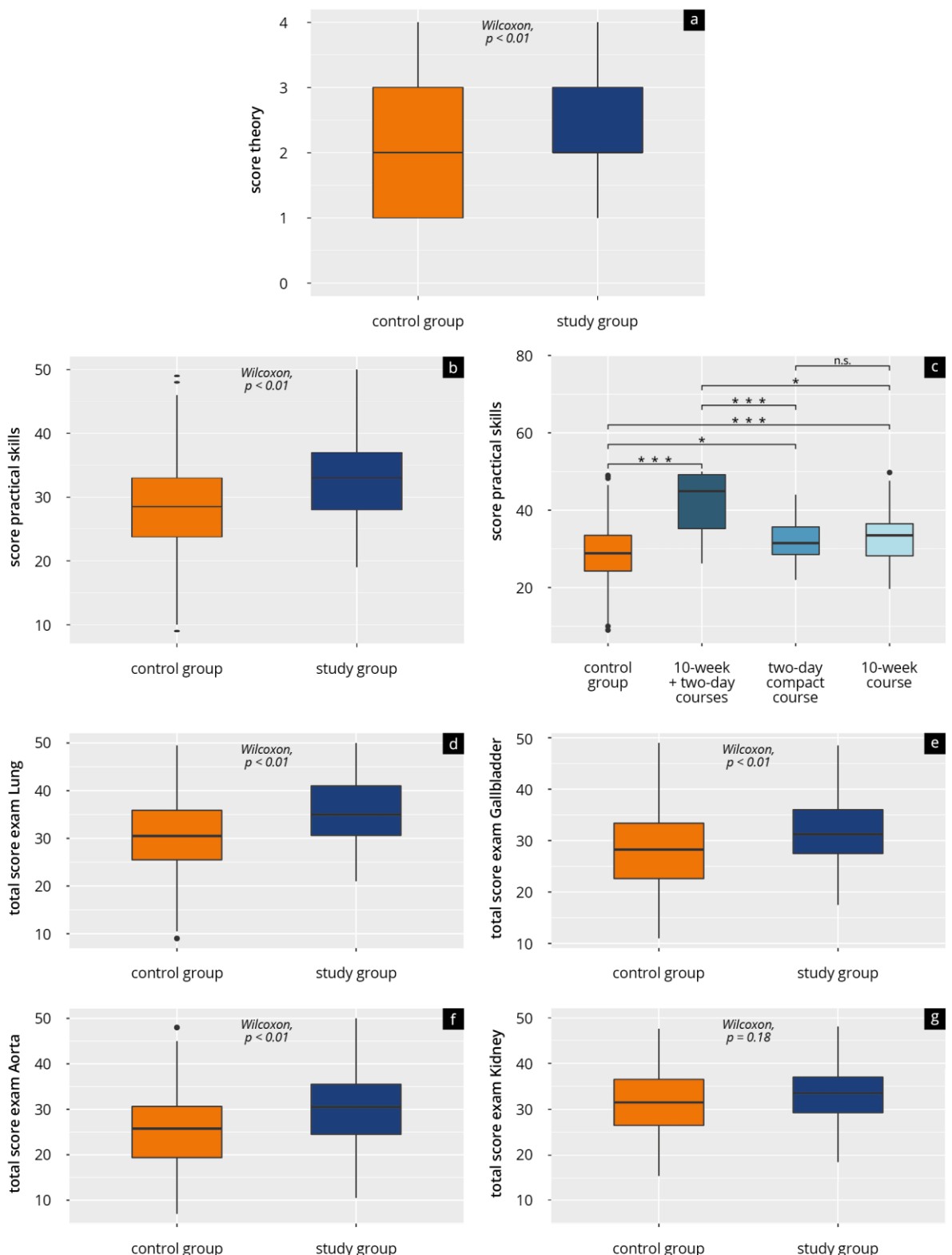

**Figure 2.** Results of theory test and practical exams for control group (did not attend ultrasound courses) and study group (attended ultrasound courses) (**a**) total theory score; (**b**) total practical score; (**c**) total practical score in subgroups; (**d**) practical score—lung, (**e**) practical score—gallbladder; (**f**) practical score—abdominal aorta; (**g**) practical score—kidney; n.s. = $p > 0.05$; * = $p \leq 0.05$; *** = $p \leq 0.001$.

## 4. Discussion

As yet there are only few data examining the effectiveness and long-term outcome of peer-assisted training approaches in ultrasonography [27–29,31,32]. The results of this prospective study suggest that integrating peer-assisted training programmes that are specific to ultrasound into the (early) clinical part of medical studies can create sustainable skills acquisition in the long term. At the start of their practical year, the participants have better practical and theoretical ultrasound skills than a comparator group who attended no ultrasound courses during their studies. Their subjective assessment of competency underlines this trend. Furthermore, the results show the students' wish to integrate digitally assisted ultrasound education programmes increasingly within compulsory teaching.

### 4.1. Discussion of Subjective Assessment of Competency

The fact that students assess their competency as better immediately after participating in ultrasound-specific training programmes has already been demonstrated in several pre/post design studies [18,19,27,39]. A steep subjective learning curve can often be measured [16,27,39] and also maintained over a period of a few months [32]. However, the actual acquisition of skills is sometimes underestimated [32]. The data recorded in our study suggest that subjective assessments of competency are higher than in the control group lasting over a longer period of time (>1 year to 2.5 years). Our measured (long-term) assessments of competency are admittedly at a slightly lower level than competency assessments requested immediately after course attendance [16,18,27,39]. This also applies to the recorded assessments of proficiency in comparative studies (8 months later) [28]. By contrast, the subgroup of those who attended several courses distributed within their medical studies assessed their current ultrasound competency at a similarly high level [28]. One explanation might be the longer interval until the follow-up observation in our study and the more homogeneous study group (same semester). This provides grounds for embedding more ultrasound contact points within compulsory teaching [10,15,25].

The greater self-confidence in one's own examination skills thereby achievable might thus lead to an easier, more stress-free entry into clinical practice. Transferability to other (specialist) disciplines or skills (for example, physical examination) also seems likely [40].

### 4.2. Discussion of Objective Measurement of Competency and Influencing Factors

It is known that by participating in (peer-assisted) ultrasound-specific education programmes, attendees can build up theoretical and practical skills that are directly and objectively measurable [9,16,18,20,21,23,27,39]. An important aspect of these education programmes is not just the short-term gain in competency but also the possible long-term learning success [27–29,31,32]. The data from our study imply that such a learning success is possible even up to 2.5 years after completion of ultrasound courses compared with a control group. Comparative studies have already demonstrated a difference in the long-term learning success of "trained" and "untrained" [27,41]. However, these studies mostly used a comparator group from lower semesters [41], no comparator group [27,31,32], and/or a very small study group [27] and chose a shorter follow-up observation period [27,41]. Furthermore, the particular studies evaluated different subject areas with different examination formats [27–32]. In our study, examination of practical and theoretical skills covered a relatively wide spectrum comprising of imaging and assessing the aorta, the kidneys, the gallbladder, and the lungs. In addition, we evaluated the medical dialogue during the ultrasound examination. Only the "kidney" practical exam showed equivalent results that were not group dependent (study group vs. control group). A possible explanation is that ultrasound use might be more frequent in the specialist disciplines of urology, gynaecology, and general medicine during university education or clinical placements. The equivalent competencies of the two groups in "communication" might be attributable due to the successful implementation of the subject "medical dialogue" in the first clinical semester. This should be investigated in more depth in further follow-up studies.

The subgroup of students who attended several ultrasound courses within their medical studies tended to achieve the best results in the theoretical and practical testing, which further supports establishing multiple points of contact with ultrasound.

The "number of independently performed ultrasound examinations of more than 50" is a major influencing factor that is not group-dependent. For example, medical certification (via DEGUM, for instance) also requires proof of a certain number of examinations as quality standards. Such a quality criterion should also be established in future student education. This might result in the coordination of uniform learning objectives but also better comparability of training curricula. The practical exams in this study only took place on healthy subjects. Examination of patients with pathological findings was not part of the testing, which should be borne in mind in future studies.

There is a significant difference between the two groups in the subjective self-assessment of theoretical knowledge—but not in the theoretical questions for all four complexes. The students may have understood the question in terms of the full theoretical competence of ultrasound. The preparation for the 2nd state examination, which took place shortly before as a theoretical examination, could provide an explanation. The content of this exam was mainly concerned with imaging techniques and pathological presentation.

### 4.3. Discussion of the Attitude to Ultrasound Teaching within Medical Studies

The existing high demand for integration of ultrasound-specific training programmes into medical education has already become clear through international surveys and review articles [3–6,8,10,14,15,24]. The evaluation data recorded in our study also confirm that students want to receive instruction in ultrasound-specific skills during their studies or during compulsory teaching. This coincides with recommendations of the National Competency-Based Learning Objective Catalogue of Medicine (NKLM) [42] and international professional associations [11–13]. Furthermore, the participants advocated greater use of digital teaching media and teaching methods within ultrasound education. It follows therefore that training approaches developed or applied in the future should increasingly be implemented on a "blended learning" basis [11–13,43–45]. Better digitally assisted preparation might make hands-on sessions (training on the equipment) more effective, which in turn could lead to better and more sustainable skills build-up. Independent practice on volunteers/patients should additionally be used following/during a training programme in order to consolidate and enhance skills [18,21].

### 4.4. Future Prospects for Curriculum Design

Ultrasound-specific training programmes are still not offered at most university locations, either as extracurricular courses or as part of compulsory teaching. Our results indicate that early implementation may motivate students to take further courses, which potentially influence curriculum choices. We implemented our training courses mainly in an early stage of a conventional degree programme. What possible advantages and disadvantages might arise should therefore be discussed. The advantage of early implementation is that it paves the way for long term self-study with consequent deepening and consolidation of ultrasound knowledge. For this purpose, clinical placements with add-on practical use of ultrasound would be welcome and could be implemented according to information provided by our participants (many came into contact with ultrasound during their clinical placements). The spacing effect thereby utilized [46,47] might thus result in more sustainable learning success. One disadvantage lies in the possibility of students forgetting what they have learnt more quickly if there is no reinforcement of the subject matter. This would result in a poor level of competency at the start of their careers. If implementation takes place at a later stage within their studies, the opposite conclusions would apply. Follow-up studies should tackle this question, which cannot be answered by the study presented here. Comparison of traditional degree programmes with so-called model degree programmes in relation to the development of ultrasound skills should also become part of future multicentre studies. These model programmes offer the

advantage of horizontal and vertical integration of learning content and associated topic and organ-centred modularisation [15,25,48].

Irrespective of the above and in summary, interdisciplinary coordination of medical specialist disciplines should continuously create multiple contact points with ultrasound [10–15,25,31]. Bearing in mind the universal lack of resources and teaching staff, good strategic planning and the employment of medically supervised peer tutors are imperative [17,22,27,49]. Modern teaching methods (for example, simulator-based training) and adapted, validated test formats [26,50–52] should also be included in future curriculum design [45,50]. This kind of longitudinally structured curriculum might thus lead to better skills development for all students in the long run. This might facilitate more effective entry into specialist training and lead to improved patient care over the long term. Future studies should therefore look at the extent to which such a curriculum brings about increased use of ultrasound by the participants and what influence this has on possible involvement in professional associations (such as DEGUM) or participation in certified medical training courses. The latter should be properly adapted to modern trends [45]. Furthermore, strengthening of national and international university exchange programmes is needed, and the aim should be to achieve certification of university ultrasound training curricula by professional associations. Being guided by medical education programmes that are already certified [39] might make this process easier. From our point of view, the implementation of a possible curriculum can look as follows: Building up basic skills in the context of the anatomy course (cross-sectional image understanding) in semesters 1–4. Basic sonography course at the beginning of the 5th semester. Deepening of sonography skills within the framework of clinical clerkships and clinical compulsory internships as well as through free practice opportunities in a skills lab. Implementation of at least two teaching units of sonography training per clinical discipline in compulsory studies. Refresher format before the start of the practical year. This process should be supported by the use of digital teaching media such as e-learning and simulators.

### 4.5. Limitations

Limitations include the question about the number of independently performed examinations; the answers were grouped, hence preventing differentiated analysis based on exact numbers. The same grouping issue applies to the question about the timing of previous course attendance. In addition, the subgroups were too small, so that we were only able to describe trends within these subgroups. In relation to the practical examination, it is important to note some of the different point distributions of the subitems, which made a direct comparison within the items impossible. Furthermore, participants each completed only two of the four possible exams. The lack of a priori randomisation is another limitation. In addition, ultrasound experience through possible curricular courses was not asked about in more detail. However, the training curriculum at the university hospital did not change significantly during the study period. We also cannot completely rule out possible selection effects and the influence of motivation. As described above, the examination was conducted on different actor patients with similar body mass indexes. Nevertheless, possible small anatomical differences could have made the examination easier or more difficult.

### 4.6. Conclusions

In conclusion, this study showed that the integration of peer-tutor-assisted ultrasound courses in the early phase of the clinical part of medical school can result in a sustainable, long-term (1–2.5 years) subjective and objective gain in competency compared with a control group. Future curriculum designs should address this aspect and embed ultrasonography ubiquitously within compulsory training through modern, digitally assisted teaching approaches.

**Supplementary Materials:** The following supporting information can be downloaded at: https://www.mdpi.com/article/10.3390/tomography9040104/s1, Figure S1: Flow chart diagram of the study population (according to CONSORT); complete handling of the written tests and participation in the practical exam were the defined inclusion criteria; Figure S2: Theory test to check pathology understanding; Supplementary 3: Examination task (lung, pleural effusions); Figure S3: Box plot diagram of subjective assessment of theoretical (a) and practical (b) competency.

**Author Contributions:** J.M.W., N.W., K.-U.S., P.H., H.B. (Holger Buggenhagen), K.D., J.K., N.B., A.M.W., L.A.L., M.R., H.B. (Henrik Bellhäuser), L.J.S., R.K., J.W.-M. and L.M. devised the study, assisted in data collection, participated in the interpretation of the data, and helped draft the manuscript. J.M.W., N.W., K.-U.S., P.H., R.K., J.W.-M. and L.M. carried out the data collection. H.B. (Holger Buggenhagen), K.D., J.K., N.B., A.M.W., L.A.L. and H.B. (Henrik Bellhäuser) supported the data collection efforts. J.M.W., N.W. and L.M. created all of the figures and participated in the interpretation of data. J.M.W., N.W. and L.M. performed the statistical analysis. All authors have read and agreed to the published version of the manuscript.

**Funding:** This research received no external funding.

**Institutional Review Board Statement:** After consultation with the local ethics committee of the State medical association of Rhineland-Palatinate ("Ethik-Kommission der Landesärztekammer Rheinland-Pfalz", Mainz, Germany) the need for approval for this study was waived and an ethics approval was not necessary. All procedures performed in studies involving human participants were in accordance with the ethical standards of the institutional and national research committee and with the 1964 Helsinki declaration and its later amendments or comparable ethical standards. Informed written consent was obtained from all the participants.

**Informed Consent Statement:** Not applicable.

**Data Availability Statement:** Data cannot be shared publicly because of institutional and national data policy restrictions imposed by the ethics committee since the data contain potentially identifying study participants' information. Data are available upon request from the Johannes Gutenberg University Mainz Medical Center (contact via weimer@uni-mainz.de) for researchers who meet the criteria for access to confidential data (please provide the manuscript title with your enquiry).

**Acknowledgments:** This study includes parts of the doctoral thesis of one of the authors (N.W.). We thank all participating students and lecturers for supporting our study. We would like to also thank C. Christe for their help in revising the figures.

**Conflicts of Interest:** The authors declare no conflict of interest.

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
