# Peer review of "Long-Term Effectiveness and Sustainability of Integrating Peer-Assisted Ultrasound Courses into Medical School—A Prospective Study"

_tomography, doi:10.3390/tomography9040104_

Round 1

Reviewer 1 Report

Dear authors,

thank you for the opportunity to review the manuscript. The publication deals with a very current and relevant topic. Especially in consideration of the fact that ultrasound examinations will certainly become even more important in the next few years and should therefore also gain in importance in student teaching.

Please allow me to make a few comments on the current manuscript:

- The introduction is short but sufficient in content. The research question is also adequately described. Perhaps the DEGUM curriculum could be described in more detail or linked as literature. Or it could be briefly explained in more detail why the focus is on the sub-areas of "abdominal ultrasound" and "emergency ultrasound".

- In the context of the methods, it would be interesting to know which curricular courses on the topic of ultrasound all students completed at what time. Because there will certainly have been some courses here (even if the comparison refers to the extracurricular offers).

- It would also be nice if the training of peer tutors could be described in more detail (what is the amount, what is the content).

- In the context of methods, it would be good to know which ultrasound devices were used for the examination. Were the machine models identical during the training and the examination?

- In the context of prior knowledge, contact with ultrasound in the field of internal medicine, GP, radiology, gynaecology and urology was queried. What about other subjects such as surgery, anaesthesiology or paediatrics? Were these deliberately excluded? Because here, as well, an influence would certainly be expected?

- Perhaps you could evaluate again why the subjective assessment of the students in terms of competence is rather low on a 7-point scale (in both groups rather in the "low range" - see table 2).

- How do you interpret the aspect that there is a significant difference between the two groups in the subjective self-assessment of theoretical knowledge - but not in the theoretical questions for all 4 complexes of the test?

- In figure 2(g) I would recommend that the scale of the Y-axis is shown identically to the other graphs 2(d,e,f). This would, in my view, facilitate visual comparability.

- Were the acting patients per examination station the same for all students or were they changed? Could differences in the conditions, such as severe obesity or similar, have influenced the results?

- The establishment of a longitudinal ultrasound curriculum seems logical as a consequence of the results and is now being practiced at more and more universities. It would be nice if the authors could outline roughly what such a curriculum should look like, taking into account the study results. From when to when? Interdisciplinary with which subjects? Integration of e-learning? Individual or repeated examinations?

- In the supplement (Figure 1), 3 participants are excluded due to non-matching inclusion criteria. What is meant by this? Perhaps the inclusion and exclusion criteria should also be described in more detail in the manuscript itself.

Thank you very much for answering my questions and I would like to wish you all the best for the revision of the manuscript!

In line 44, a single space should be added before "12".

Author Response

Please see the attachement below!

Reviewer 2 Report

Present manuscript highlights the potential and need for standardised implementation of ultrasound courses in the medical education curriculum. Several comments given.

1.      The novelty in the current article by the authors is too weak. The past has seen extensive published work of written material. It is required to provide more details for more explanation about the present novel in the introductory section.

2.      Line 37, please make it into passive, do not use “we”.

3.      Line 57, please explain several methods for diagnostic first before specific to ultrasound.

4.      Line73, so what is the novel for the problems brings? It is like a case study.

5.      Line 89, the basis of participate selection, criteria, and procedure needs more enhancement.

6.      Line 104, the Figure 1 needs more explanation.

7.      Line 125, apart from supplementary. The basic information would be given.

8.      Apart from ultrasound, please explain imaging also have been widely used in medical application.

9.      Please explain potential further study in silico/computational simulation in medical field. It bring several advantages such as lower cost and faster results.

-

Author Response

Please see the attachment below!

Round 2

Reviewer 1 Report

Dear Authors,

Thank you for sending me the revised manuscript and the opportunity to review it again. In my opinion, the revisions have once again improved the manuscript and I have no further comments.

The only thing I would recommend is to add the bracket "0 (0.0)" to Table 1 for the now added subjects to make it consistent.

I have not yet fully understood the reasoning regarding my comment about Figure 2. I would leave Figure 2c as it is. However, I would construct the y-axis of figure 2g in the same way as figures 2d, 2e, 2f. Then I think the comparability would also be better - the p-value can be left as it is. But ultimately, of course, this is a decision for the authors.

Thank you very much!

I would add the brackets in Table 1 as described above.

Author Response

Please see the attachement below!

Reviewer 2 Report

Reviewers greatly appreciate the efforts that have been made by the author to improve the quality of their articles after peer review. I reread the author's manuscript and further reviewed the changes made along with the responses from previous reviewers' comments. Unfortunately, the authors failed to make some of the substantial improvements they should have made making this article not of decent quality with biased, not cutting-edge updates on the research topic outlined. In addition, the author also failed to address the previous reviewer's comments, especially on comments number 1 (lack of novel), 8 (not incorporated), and 9 (not incorporated). Thank you very much for the opportunity to read the author's current work. 

-

Author Response

Please see the attachement below!
